# Intergenerational and Intersexual Differentiation in Respiratory Metabolic Rates of *Schlechtendalia chinensis*: A Comparison Across Sexuales, Parental Sexuparae, and Progeny Fundatrices

**DOI:** 10.3390/insects16101015

**Published:** 2025-10-01

**Authors:** Shuxia Shao, Bo Jiang, Xin Xu, Zhaohui Shi, Chang Tong, Zixiang Yang

**Affiliations:** 1Yunnan Key Laboratory of Breeding and Utilization of Resource Insects, Key Laboratory of Protection and Utilization of Insects, National Forestry and Grassland Administration, Institute of Highland Forest Science, Chinese Academy of Forestry, Kunming 650224, China; shuxiashao@126.com (S.S.); xuxin1995614@163.com (X.X.); zhaohui1019847069@163.com (Z.S.); tongchang186@163.com (C.T.); 2Southwest Academy of Inventory and Planning, National Forestry and Grassland Administration, Kunming 650031, China; 15887221239@163.com

**Keywords:** *Schlechtendalia chinensis* (Bell), sexuale, sexuparae, fundatrix, intergenerational and intersexual differentiation, respiratory metabolic rate

## Abstract

**Simple Summary:**

The sexual generation of *Schlechtendalia chinensis* is critical for gallnut production yet cannot feed due to complete mouthpart degeneration. This constraint presents fundamental questions concerning their energy homeostasis: Could modulation of the respiratory metabolic rate serve as an evolutionary adaptation to compensate for nutritional deficits? This study quantified respiratory metabolic rates across key developmental stages of the sexual morphs, encompassing their parental sexuparae and progeny fundatrices. All morphs exhibited lower nocturnal than diurnal respiratory metabolic rates. Males showed respiratory metabolic rates 2–3 times higher than those of females or sexuparae, with those of fundatrices being intermediate. Both sexes displayed a distinct metabolic trajectory: respiratory metabolic rates were elevated at birth, declined initially, peaked at sexual maturity (day 8, coinciding with mating), then sharply declined post-mating. This investigation addresses critical knowledge gaps in the respiratory metabolism of *S. chinensis* while elucidating an evolutionary adaptation strategy for nutrient allocation through respiratory metabolic regulation under obligate non-feeding conditions. These findings provide actionable insights for optimizing gallnut production within controlled cultivation systems.

**Abstract:**

The sexual generation of *Schlechtendalia chinensis* (Bell) is pivotal for gallnut yield yet cannot feed due to mouthpart degeneration. Could respiratory metabolic rate (RMR) modulation compensate for nutritional deficits? We quantified the RMR across key developmental stages of sexual morphs (including parental sexuparae and progeny fundatrices) using an LI-6400XT portable photosynthesis system equipped with a customized insect respiration chamber (6400-89). All morphotypes exhibited significantly lower nocturnal RMRs compared to their diurnal rates (*p* < 0.05), while RMRs did not differ significantly between morning (9:00–12:00) and afternoon (14:00–17:00) (*p* > 0.05). Significant RMR variation occurred among morphotypes: females and sexuparae displayed the lowest rates, fundatrices were intermediate, and males exhibited remarkably elevated rates (2–3 times higher than those of females or sexuparae). Both sexes showed a characteristic RMR trajectory: elevated at birth and declining during early postnatal development, followed by a gradual resurgence that culminated in peak values on postnatal day 8, coinciding with mating. This physiological zenith was immediately succeeded by marked respiratory metabolic downregulation following copulation, with RMRs decreasing substantially during the post-copulatory phase. Our findings demonstrate significant intergenerational and intersexual RMR differentiation. This research addresses critical knowledge gaps in the respiratory metabolism of *S. chinensis*, is the first to elucidate a nutrient adaptation strategy through respiratory metabolic regulation under non-trophic conditions, and provides actionable insights for optimizing gallnut production in controlled cultivation systems.

## 1. Introduction

Respiratory metabolism constitutes a fundamental physiological mechanism in insects, functioning as the principal pathway for energy transduction. This biochemical process involves the oxidative breakdown of intracellular energy substrates, primarily carbohydrates and lipids, through oxygen-mediated catabolic reactions to generate adenosine triphosphate (ATP) and concomitantly release carbon dioxide as a metabolic byproduct [1,2]. The regulation of insect respiratory metabolism operates through complex interplay among endogenous physiological determinants and exogenous environmental parameters. The key modulatory factors include organismal characteristics (body mass, developmental phase, and species-specific behavioral patterns) and ambient conditions (thermal regime, humidity gradients, atmospheric oxygen tension, and photoperiodic cycles) [3,4,5,6,7,8]. Notably, under nutritional deprivation or environmental stressors, various insect species demonstrate adaptive metabolic depression characterized by significant downregulation of respiratory rates. This hypometabolic state facilitates conservation of energy reserves, thereby enhancing survival capacity during prolonged adverse conditions [4,7,9,10].

*Schlechtendalia chinensis* (Bell) is the gall-forming aphid. Its horned galls induced on *Rhus chinensis* constitute approximately 70–75% of the total national gallnut yield, representing the principal determinant of commercial gallnut yield [11,12]. The life cycle of *S. chinensis* is complex. Each year in late March, newly emerged sexuparae (spring migrants) migrate from their winter host (mosses) to their summer host, *R. chinensis*. They reproduce, forming sexuales (females and males) in cracks of the host tree bark. These sexuales undergo three to four molts to reach maturity, after which mating occurs. Within one week post-mating, the males die, while the females continue development for approximately 20 days before producing the fundatrices. Each fundatrix relocates to the leaf tissues of *R. chinensis*, where its feeding activity induces the formation of a gall. Inside the gall, it reproduces parthenogenetically, yielding three successive generations of fundatrigeniae. The first two generations consist of apterous (wingless) fundatrigeniae, while the third generation are alate (winged) fundatrigeniae. Between late September and early October, the mature galls dehisce, releasing the alate fundatrigeniae, which then return to mosses—the winter host. There, they deposit overwintering nymphs. These overwintering nymphs undergo development and emerge the following spring (late March) as sexuparae, thereby completing the annual cycle. Thus, *S. chinensis* exhibits a heteroecious holocyclic life history, characterized by six successive morphotypes: sexuales, fundatrices, apterous fundatrigeniae, alate fundatrigeniae (autumn migrants), overwintering nymphs, and sexuparae (spring migrants) [13,14,15]. Notably, the sexuales present unique biological constraints: the complete absence of functional mouthparts, rendering them incapable of nutrient acquisition by means of feeding [16]. Furthermore, sexuales constitute the only stage in which sexual reproduction occurs, facilitating genetic recombination through mating. This process enhances genetic diversity within the population, thereby increasing evolutionary adaptability and enabling the offspring—specifically, the fundatrices—to better respond to heterogeneous and dynamic environmental conditions. Moreover, their reproductive strategy is restricted to single-ovum sexual reproduction, with each gravid female producing merely one offspring—a fundatrix. Given that individual fundatrices typically initiate solitary gall formation, the reproductive success of sexuales fundamentally governs gallnut productivity through strict 1:1 female-to-gall numerical correspondence.

The reproductive cycle of *S. chinensis* sexuales necessitates sequential mate localization and copulatory behavior, followed by embryonic development and subsequent oviposition by gravid females. These energetically demanding processes—including mate-seeking locomotion, gametogenesis, and parturition—require substantial nutrient allocation. Paradoxically, sexual morphs exhibit complete mouthpart degeneration, eliminating feeding capacity throughout their whole life. This physiological constraint raises critical questions regarding their energy homeostasis: Could metabolic rate modulation, particularly respiratory downregulation, serve as an evolutionary adaptation to offset nutritional deficits during sexual reproduction? Current entomological research lacks empirical data on respiratory metabolism dynamics in *S. chinensis*, with no published studies quantifying its catabolic processes or energy budgeting strategies during critical life stages.

This investigation employed an LI-6400XT portable photosynthesis system with a customized insect respiration chamber (6400-89) to quantify respiratory metabolic rates (RMRs) at different developmental stages of *S. chinensis* sexual morphs, including their parental sexuparae and progeny fundatrices. Utilizing high-resolution respirometry, we systematically evaluated the hypothesis that sexual aphids mitigate endogenous nutrient depletion through respiratory depression, while elucidating their evolutionary metabolic trade-offs. These findings establish mechanistic links between hypometabolic adaptation and reproductive success, providing actionable insights for optimizing gallnut production in controlled cultivation systems.

## 2. Materials and Methods

### 2.1. Insect Materials

Field-collected specimens of *S. chinensis* were obtained from the designated area for gallnut cultivation (30°10′12″ N, 110°52′36″ E; elevation 960 ± 15 m) in Bainianguan, Wufeng Tujia Autonomous County, Hubei Province. Newly emerged sexuparae were collected from the cultivated moss nursery and subsequently transferred to 90 mm Petri dishes lined with moistened filter paper under controlled laboratory conditions (25 ± 1 °C, 80% RH, 16:8 L:D). Continuous monitoring commenced upon initiation of parturition, with daily collection of neonate progeny (males and females) and the newborn fundatrices produced by females.

### 2.2. RMR Measurements

We employed an LI-6400XT portable photosynthesis system (LI-COR Biosciences, Lincoln, NE, USA)) equipped with a customized insect respiration chamber (6400-89) to quantify respiratory metabolic rates across three distinct daily phases: 9:00–12:00, 14:00–17:00, and 19:00–22:00. The experimental design incorporated four aphid morphotypes (sexuparae, females, males, and fundatrices), with all specimens selected for uniform size and confirmed vitality through pre-experimental screening. Prior to measurements, 100 sexuparae or 400 individuals for other morphotypes underwent 30 min of environmental acclimation within the respiration chamber under standardized conditions (24 ± 0.5 °C, 70 ± 5% RH). Continuous CO_2_ flux monitoring commenced following stabilization of the emission curves (defined as <5% variation over 5-min intervals), with high-frequency data acquisition at 2-s resolution. Each group of samples was measured for 15 min, and the respiratory metabolic rate (μg·g^−1^·min^−1^) was automatically recorded by the instrument recorder. The experimental design implemented five independent biological replicates per morphotype–temporal combination, with strict avoidance of individual reuse across the experimental groups.

### 2.3. Statistics and Analysis

Initial data handling was performed in Microsoft Excel 365. Statistical analyses were performed in R (v4.5.0; R Development Core Team, Vienna, Austria) using the “stats” package, including outlier detection (Tukey fence) and calculations of descriptive statistics (mean ± SD, median [IQR]). Visualizations were completed in R using the ggplot2 package(v4.0.0). The normality of the respiratory metabolic rate (RMR) variable was formally evaluated with the Shapiro–Wilk test (*n* = 3995). The statistic was significant (W = 0.912, *p* < 0.001), indicating departure from normality; visual inspection revealed a pronounced right-tail skew (Figure 1). Consequently, inter-group comparisons were conducted using non-parametric tests. The Kruskal–Wallis rank-sum test was chosen as the primary inferential procedure because it accommodates multiple independent samples without assuming normality and is robust to outliers. When the Kruskal–Wallis test indicated a significant difference among groups (*p* < 0.05), post hoc pairwise comparisons were performed with Dunn’s test, applying Bonferroni correction to control the family-wise Type I error rate and to pinpoint the specific pairs driving the overall effect. This analytical approach enabled comprehensive comparisons across two dimensions: inter-morphotype differences among the 4 aphid morphotypes and intra-morphotype variations across the 3 temporal intervals. The experimental design specifically addressed both interspecific metabolic divergence and temporal metabolic fluctuations within identical morphotypes, ensuring a rigorous evaluation of statistical significance across all comparative groups.

## 3. Results

### 3.1. Analyses of Respiratory Metabolic Rates Across Four S. chinensis Morphotypes

#### 3.1.1. Intra-Morphotype Variations in Respiratory Metabolic Rates Across Three Temporal Intervals in a Day

As shown in Figure 2, the Kruskal–Wallis test revealed highly significant differences in respiratory metabolic rates (RMRs) among the three time intervals for both the males and fundatrices (*p* < 0.001). In contrast, no significant temporal variation in the RMR was detected for either the sexuparae or females (*p* = 0.056 and *p* = 0.846, respectively).

Subsequent pairwise comparisons with Dunn’s test (Bonferroni correction) showed that the males’ nocturnal median RMR (212.4 μg·g^−1^·min^−1^) was significantly lower than both their morning (239.0 μg·g^−1^·min^−1^, *p* < 0.05) and afternoon values (235.7 μg·g^−1^·min^−1^, *p* < 0.05), whereas the difference between morning and afternoon remained non-significant (*p* > 0.05).

Dunn’s post hoc comparisons (Bonferroni-adjusted) further resolved the temporal patterns for the fundatrices: their nocturnal median RMR (114.3 μg·g^−1^·min^−1^) was markedly reduced relative to both their morning (170.8 μg·g^−1.^ min^−1^, *p* < 0.001) and afternoon values (178.8 μg·g^−1^·min^−1^, *p* < 0.001), whereas the morning and afternoon RMRs did not differ significantly (*p* = 0.702). The females maintained a stable RMR (~104 μg·g^−1^·min^−1^) across the three time intervals, with no detectable fluctuation (*p* = 0.846).

The sexuparae exhibited only a modest decline from afternoon (94.0 μg·g^−1^·min^−1^) to night (72.4 μg·g^−1^·min^−1^), and the differences in the RMR among the three time intervals We confirm that the spaces have been replaced with center dots throughout the manuscript as requested were not statistically significant (*p* = 0.056).

#### 3.1.2. Inter-Morphotype Differences in RMRs Among Four Morphotypes of *S. chinensis*

This study provides a comprehensive comparison of RMRs among four *S. chinensis* morphotypes (Figure 3). Kruskal–Wallis tests revealed highly significant temporal heterogeneity in RMRs among the four examined morphotypes of *S. chinensis* (sexuparae, females, males, and fundatrices; all *p* < 0.001). Dunn’s post hoc comparisons with Bonferroni correction further identified the following pairwise differences:

Males demonstrated superior respiratory activity across all three temporal intervals. Specifically, the male median RMRs measured 239.047, 235.694, and 212.385 μg·g^−1^·min^−1^ during the morning (09:00–12:00), afternoon (14:00–17:00), and evening (19:00–22:00) periods, respectively, significantly exceeding those for the other three *S. chinensis* morphotypes (*p* < 0.05 for all phases). The fundatrices exhibited secondary respiratory metabolic activity, with median rates of 170.824, 178.819, and 114.347 μg·g^−1^·min^−1^ across corresponding time intervals, representing a 24.131–46.161% reduction relative to the rates for males (*p* < 0.05 for all phases). Notably, the female median RMRs were markedly attenuated, registering only 43.689–448.315% of the male values (104.437, 104.372, and 102.613 μg·g^−1^·min^−1^; *p* < 0.05 for all phases), thereby underscoring significant sexual dimorphism in respiratory metabolism. Sexuparae displayed the lowest median RMRs (85.534, 94.032, and 72.427 μg·g^−1^·min^−1^), though these values did not differ statistically from those for females (*p* > 0.05 for the three corresponding time intervals).

### 3.2. Developmental Dynamics of Respiratory Metabolic Rates in Males and Females of S. chinensis

Figure 4 delineates the RMR patterns of the female *S. chinensis*. Neonatal females (postnatal day 1) displayed relatively high respiratory metabolic activity, with median RMRs of 129.810 (morning), 132.852 (afternoon), and 144.507 μg·g^−1^·min^−1^ (evening), ranking second among the seven developmental stages. A significant median RMR decline in the three temporal intervals (81.892, 89.833, and 92.553 μg·g^−1^·min^−1^, respectively; *p* < 0.001 for each interval) accompanied sexual maturation by postnatal day 5.

Notably, a mating-phase RMR resurgence emerged during mating (postnatal days 5–8), with median RMRs progressively escalating to peak values on postnatal day 8 (192.065, 163.293, and 162.802 μg·g^−1^·min^−1^, respectively, for the three temporal intervals); the peak values were significantly higher than those for other developmental stages (*p* < 0.05), except for those on the first day in the afternoon and evening (*p* > 0.05).

Post-copulatory respiratory metabolic suppression ensued. By postnatal day 23 (pre-parturition), the morning median RMR transiently rebounded to 109.873 μg·g^−1^·min^−1^, while the afternoon (76.931 μg·g^−1^·min^−1^) and nocturnal (40.773 μg·g^−1^·min^−1^) measurements reached ontogenetic minima, suggesting energy reallocation toward viviparous preparation.

As illustrated in Figure 5, male RMRs were approximately twice those of females, with parallel dynamic patterns exhibited across both sexes. Neonatal males (postnatal day 1) demonstrated pronounced RMR values across the three temporal intervals: morning (225.045 μg·g^−1^·min^−1^), afternoon (296.643 μg·g^−1^·min^−1^), and evening (207.840 μg·g^−1^·min^−1^). Subsequently, RMRs exhibited a transient decline prior to sexual maturation. Upon entering the mating phase (postnatal days 5–8), males displayed a dramatic surge in metabolic activity, achieving maximal lifetime RMR values by day 8 (morning: 416.816 μg·g^−1^·min^−1^; afternoon: 380.319 μg·g^−1^·min^−1^; evening: 315.935 μg·g^−1^·min^−1^). These values significantly exceeded those observed during other developmental stages (*p* < 0.05 for all comparisons across developmental stages). Post-copulatory senescence was characterized by gradual RMR attenuation, though the rates were still relatively high near the end of life. Notably, even at terminal life stages (postnatal day 11), males maintained elevated RMR values (morning: 242.391 μg·g^−1^·min^−1^; afternoon: 260.772 μg·g^−1^·min^−1^; evening: 267.368 μg·g^−1^·min^−1^), demonstrating persistent metabolic investment throughout their life history.

## 4. Discussion

Insect respiratory intensity serves as a direct proxy for quantifying energy metabolism [17]. Crucially, insects modulate their respiratory intensity as a physiological strategy to enhance environmental adaptability under adverse conditions [18,19]. Consequently, analyses of respiratory metabolism patterns provide dual insights: they elucidate internal nutrient utilization dynamics and reveal evolutionary trade-offs between environmental adaptation and life-history strategies in insects. This study constitutes the first systematic investigation of respiratory metabolism across three distinct generations of *S. chinensis*—the sexual generation (males and females), their maternal generation (sexuparae), and their offspring generation (fundatrices)—while simultaneously examining sexual dimorphism within males and females. Our findings demonstrate significant intergenerational and intersexual differentiation in respiratory metabolic rates (RMRs). These variations exhibit precise functional alignment with species-specific ecological roles and environmental adaptation mechanisms.

This investigation quantitatively tracked RMR dynamics across developmental stages in both sexes, revealing similar temporal RMR patterns coupled with significant sexual RMR dimorphism. Females demonstrated stage-specific RMR regulation characterized by three distinct phases: an initial neonatal downregulation, reproductive-phase modulation, and post-copulatory depression. Neonatal females exhibited elevated RMRs, followed by 34.2–40.7% suppression within postnatal days 1–5 (*p* < 0.05), indicating strategic energy reallocation toward reproductive system maturation and long-term embryogenesis requirements. Subsequent mating phase activation (days 5–8) manifested progressive RMR elevation, peaking at 192.065 μg·g^−1^·min^−1^ on day 8—a metabolic signature associated with oocyte activation, spermathecal function, and vitellogenesis-related hormonal shifts [4]. A post-mating RMR collapse to 40.773 μg·g^−1^·min^−1^ (day 23) suggests evolutionary adaptation through extreme RMR throttling to optimize maternal longevity and fundatrix production success.

In striking contrast, male aphids exhibited high respiratory metabolic activity throughout the day and night, with their RMR being approximately twice that of females. This sexual dimorphism was particularly significant during the mating period (days 5–8), with the RMR reaching as high as 416.816 μg·g^−1^·min^−1^ on the 8th day (peak RMR). This sustained hypermetabolism is likely related to mate-searching behaviors, courtship displays, and intrasexual competition [3,20], with late-life maintenance of elevated RMRs (242.391–267.368 μg·g^−1^·min^−1^) potentially contributing to rapid post-mating senescence (3–5-day mortality window). The observed sexual RMR dichotomy reflects fundamental evolutionary trade-offs between female resource allocation strategies and male expenditure-driven reproductive tactics.

Under conditions of exogenous nutritional deprivation, females, as exclusive carriers of embryonic development, implement prioritized energy allocation through dynamic regulation of respiratory metabolism to ensure fundatrix embryogenesis. This adaptive mechanism aligns with the resource investment paradigm characteristic of K-selected species. Notably, the significant reduction in post-mating RMRs exemplifies their evolutionary “metabolic thrift” strategy, wherein deep RMR suppression extends nutrient reserve utilization cycles to guarantee complete fundatrix development. In contrast, males demonstrate respiratory metabolic patterns congruent with r-selected strategy parameters, emphasizing rapid metabolic turnover and short-term reproductive efficiency. This fundamental sexual dimorphism in metabolic regulation corresponds to the core premise of sexual selection theory: differential reproductive investment strategies, with males optimizing mating competition capacity and females prioritizing offspring provisioning [4]. The observed dichotomy in respiratory physiology—sustained metabolic economy in females versus transient metabolic intensity in males—provides mechanistic evidence for sex-specific life-history strategies shaped by distinct evolutionary pressures. This metabolic specialization enables females to maintain developmental homeostasis during prolonged nutritional challenges, while males maximize immediate reproductive output through rapid energy mobilization.

This study revealed significant intersexual and intergenerational variations in RMRs among *S. chinensis* populations. Notably, males demonstrated the highest metabolic activity across diurnal and nocturnal cycles (the average value for their whole developmental stage was 212.385–239.047 μg·g^−1^·min^−1^), exhibiting statistically significant elevation compared to other morphotypes of *S. chinensis.* The RMR hierarchy followed a distinct pattern: fundatrices ranked second (114.347–178.819 μg·g^−1^·min^−1^), followed by females (about 104.372 μg·g^−1^·min^−1^), with sexuparae displaying the lowest metabolic rates (72.427–94.032 μg·g^−1^·min^−1^). This RMR pattern appears to be evolutionarily conserved through functional adaptation. The heightened respiratory metabolic demand in males correlates with energy-intensive reproductive behaviors, including mate location and copulatory activities. Conversely, females maintain reduced respiratory metabolic states to optimize energy allocation for sustained oviposition and embryonic development. Fundatrices’ elevated RMRs likely facilitate host plant colonization through enhanced locomotor capacity. The respiratory metabolic suppression observed in sexuparae suggests dual adaptive strategies: (1) the conservation of energetic reserves for migratory flight requirements and (2) the extension of reproductive longevity through RMR modulation, thereby ensuring sufficient production of sexual offspring [20].

Circadian analysis of the RMRs revealed conserved diurnal oscillation patterns across all four aphid morphotypes, characterized by elevated diurnal and reduced nocturnal respiratory metabolic activity. However, quantitative differentials emerged: fundatrices exhibited statistically significant nocturnal metabolic suppression compared to their diurnal levels. This differential regulation appears to be evolutionarily conserved through photoperiod-mediated physiological modulation. The observed RMR downregulation corresponds with reduced nocturnal host-seeking behaviors. Such chronobiological adaptation likely optimizes energy conservation strategies while maintaining essential physiological functions under suboptimal environmental conditions [21].

This investigation addresses critical knowledge gaps regarding the respiratory metabolism of *S. chinensis* while elucidating its evolutionary adaptation strategy for nutrient allocation through respiratory metabolic regulation under non-trophic conditions. Our findings establish this species as a novel experimental system for studying extreme environmental adaptation mechanisms in insects, providing a theoretical foundation for optimizing sexual aphid conservation protocols in gallnut cultivation systems. However, this study did not delineate the molecular architecture of respiratory metabolic pathway regulation in *S. chinensis*—a critical knowledge gap that could be elucidated through integrated metabolomic profiling and flux analysis. Furthermore, the influences of environmental factors (such as thermal and hygric) on the RMRs of sexuales require systematic evaluation via controlled laboratory simulations, which would enable environmental parameter optimization and refinement in artificial rearing systems.

## 5. Conclusions

This study provides the first comprehensive analysis of respiratory metabolic rate (RMR) dynamics across three distinct generations of *S. chinensis* (Bell): the sexual generation and their parental sexuparae and progeny fundatrices. Our findings demonstrate that there are significant intergenerational and intersexual RMR differentiations. This RMR pattern aligns precisely with ecological roles: elevated RMRs in males support intense mating activity, intermediate RMRs in fundatrices facilitate host colonization, and suppressed RMRs in sexuparae and females conserve energy for migration and sustained sexual offspring production. RMR modulation in sexuales serves as a critical evolutionary adaptation to compensate for nutritional deficits caused by complete mouthpart degeneration.

## Figures and Tables

**Figure 1 insects-16-01015-f001:**
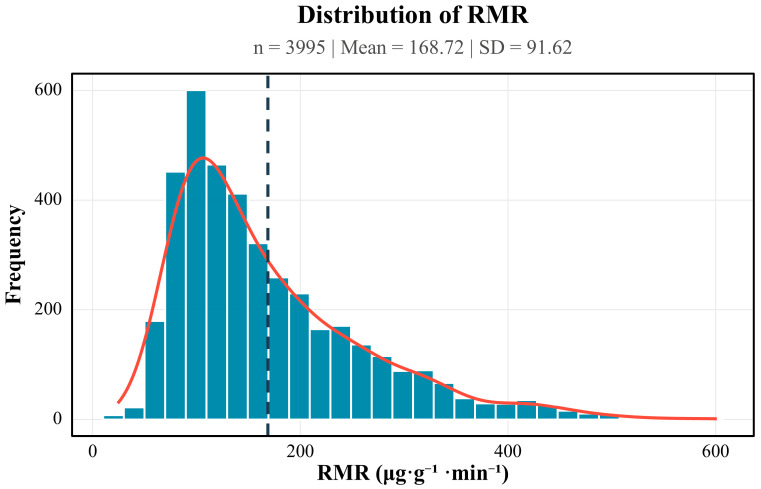
Frequency distribution of respiratory metabolic rates (RMRs) in four morphotypes of *S. chinensis.* The blue bar chart displays the frequencies of values across different RMR intervals, while the red curve represents the fitted line, collectively indicating a skewed distribution of the data.

**Figure 2 insects-16-01015-f002:**
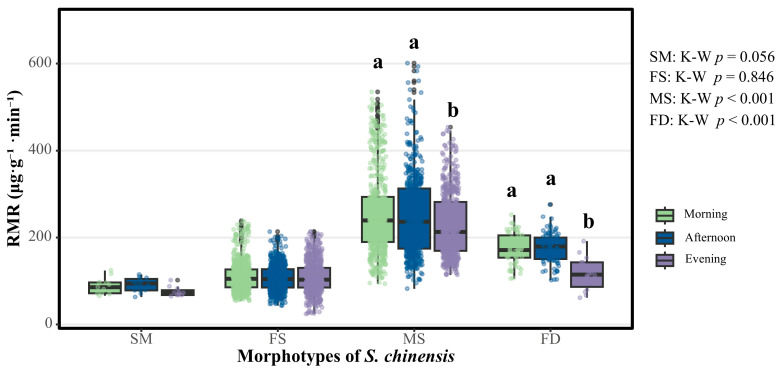
Circadian variations in respiratory metabolic rates for four morphotypes of *S. chinensis*. Abbreviations: SM, sexuparae; FS, females; MS, males; FD, fundatrices. Boxes depict medians and inter-quartile ranges; whiskers extend to the 5th and 95th percentiles. Kruskal–Wallis test: SM, *p* = 0.056; FS, *p* = 0.846; MS, *p* < 0.001; FD, *p* < 0.001. Different lowercase letters (a, b) indicate significant differences within a morphotype after Dunn’s post hoc test with Bonferroni correction (α = 0.05).

**Figure 3 insects-16-01015-f003:**
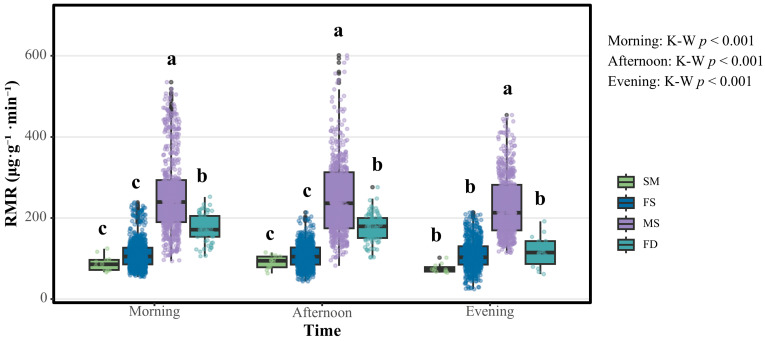
A comparison of RMRs among four *S. chinensis* morphotypes at three daily timepoints. Abbreviations: SM, sexuparae; FS, females; MS, males; FD, fundatrices. Boxes show medians and inter-quartile ranges; whiskers indicate 5th–95th percentiles. Kruskal–Wallis test: morning, afternoon, and evening; all *p* < 0.001. Different lowercase letters (a, b, c) indicate significant differences among morphotypes within each time interval (Dunn’s post hoc with Bonferroni correction, α = 0.05).

**Figure 4 insects-16-01015-f004:**
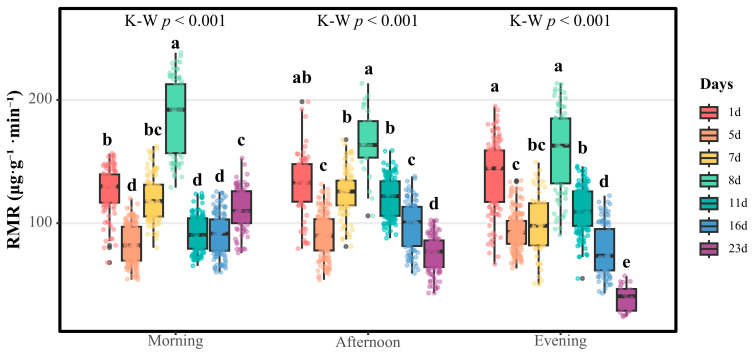
Respiratory metabolic rates of female *S. chinensis* at different developmental stages. Boxes represent medians ± inter-quartile ranges; whiskers extend to 5th–95th percentiles. Kruskal–Wallis tests: morning, afternoon, and evening; all *p* < 0.001. Different lowercase letters (a–e) indicate significant differences among developmental stages within each time interval (Dunn’s post hoc with Bonferroni correction, α = 0.05).

**Figure 5 insects-16-01015-f005:**
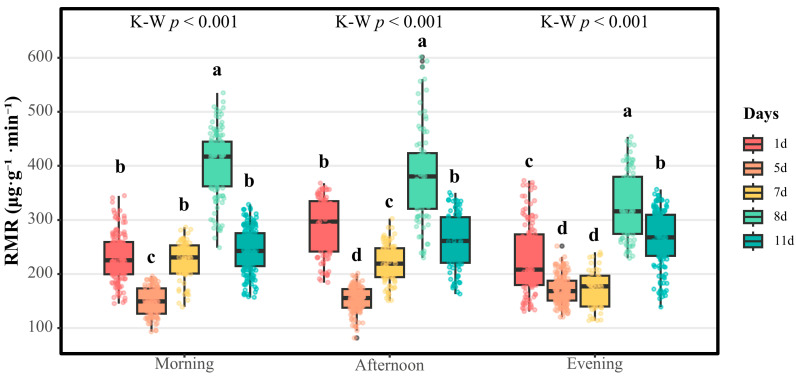
Respiratory metabolic rates of male *S. chinensis* at different developmental stages. Boxes represent medians ± inter-quartile ranges; whiskers extend to 5th–95th percentiles. Kruskal–Wallis tests: morning, afternoon, and evening; all *p* < 0.001. Different lowercase letters (a–d) indicate significant differences among developmental stages within each time interval (Dunn’s post hoc with Bonferroni correction, α = 0.05).

## Data Availability

Data are contained within the article.

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
