# Peer review of "Intergenerational and Intersexual Differentiation in Respiratory Metabolic Rates of Schlechtendalia chinensis: A Comparison Across Sexuales, Parental Sexuparae, and Progeny Fundatrices"

_insects, 2025, doi:10.3390/insects16101015_

Round 1

Reviewer 1 Report

Comments and Suggestions for Authors

Review

Title: Analysis of respiratory metabolic rates across three distinct generations of gall aphid, Schlechtendalia chinensis–the sexual generation and its parental and progeny generations.

This study quantified respiratory metabolic rates of the gall aphid across key developmental stages of the sexual morphs, encompassing their parental sexuparae and progeny fundatrices.

The paper is generally well written, but it has several weaknesses that has to be improved before it can be reconsidered for publication. Please see my comments below.

The title must be changed, the world ‘generations’ is 3times present. Please add a different and simplest title.

Please explain in abstract why this research is new and important

Keywords: please use different keywords as the manuscript title.

Methods: Please add the sample size for insects used for the experiment.

By using the RMR, what kind of data you detected? What was the data format, the data transformation methods before these were statistically analyzed. Where these normally distributed?

After ANOVA a Tukey or a Mann-Whitney test is used. What is the Fisher used for?

According to the results and figure 1, there were a high variations in the data, how is possible to sign with ‘a’ all the statistic non-differences or differences? I think, the data was not distributed normally, we don’t know this, as we do not get information about the quantitative data obtained, but if so, the right methods would be a Kruskal-Wallis test and a boxplot presentations where medians would have importance.

Also at figure 1, if the SE are presented above bars, and at SM and FD there are differences (a and b labels above bars, how is possible that the SE bars are overlapping? Altogether, I have serious problems with the data obtained and its data analyses.

The same problems occurred at figure 2 and its explanations. The figure 3 and 4 can not be interpreted, please use a different format, maybe boxplots.

I would rather evaluate the discussion part after the Results part clarifications.

Comments on the Quality of English Language

Review

Title: Analysis of respiratory metabolic rates across three distinct generations of gall aphid, Schlechtendalia chinensis–the sexual generation and its parental and progeny generations.

This study quantified respiratory metabolic rates of the gall aphid across key developmental stages of the sexual morphs, encompassing their parental sexuparae and progeny fundatrices.

The paper is generally well written, but it has several weaknesses that has to be improved before it can be reconsidered for publication. Please see my comments below.

The title must be changed, the world ‘generations’ is 3times present. Please add a different and simplest title.

Please explain in abstract why this research is new and important

Keywords: please use different keywords as the manuscript title.

Methods: Please add the sample size for insects used for the experiment.

By using the RMR, what kind of data you detected? What was the data format, the data transformation methods before these were statistically analyzed. Where these normally distributed?

After ANOVA a Tukey or a Mann-Whitney test is used. What is the Fisher used for?

According to the results and figure 1, there were a high variations in the data, how is possible to sign with ‘a’ all the statistic non-differences or differences? I think, the data was not distributed normally, we don’t know this, as we do not get information about the quantitative data obtained, but if so, the right methods would be a Kruskal-Wallis test and a boxplot presentations where medians would have importance.

Also at figure 1, if the SE are presented above bars, and at SM and FD there are differences (a and b labels above bars, how is possible that the SE bars are overlapping? Altogether, I have serious problems with the data obtained and its data analyses.

The same problems occurred at figure 2 and its explanations. The figure 3 and 4 can not be interpreted, please use a different format, maybe boxplots.

I would rather evaluate the discussion part after the Results part clarifications.

Reviewer 2 Report

Comments and Suggestions for Authors

The manuscript “Analysis of respiratory metabolic rates across three distinct generations of gall aphid, Schlechtendalia chinensis–the sexual generation and its parental and progeny generations” (authors Shuxia Shao, Bo Jiang, Xin Xu, Zhaohui Shi, Chang Tong and Zixiang Yang) is dedicated to quantifying respiratory metabolic rates at different developmental stages of S. chinensis sexual morphs, including their parental sexuparae and progeny fundatrices. The data presented in the MS actually fill critical gaps in knowledge about the respiratory metabolism of S. chinensis, while clarifying the evolutionary strategy of adaptation to nutrient distribution through regulation of respiratory metabolism in conditions of mandatory malnutrition. Despite the generally very favorable impression of the MS text, it is still not free from some shortcomings. First of all, in the Introduction it is worth giving a brief description of the life cycle of this heterocious aphid species, so that the reader can understand the idea of the authors: “The sexual generation of Schlechtendalia chinensis is critical for walnut production” (lines 13, 14). Further, the Simple Summary says: “These findings provide actionable insights for optimizing gallnut production within controlled cultivation systems”, and in the Abstract we read: “These variations exhibit precise functional alignment with species-specific ecological roles and environmental adaptation mechanisms, providing actionable insights for optimizing gallnut production in controlled cultivation systems”. However, these theses are completely not disclosed in the MS, which is not surprising, because the MS so far only outlines the direction of further work: “Our findings establish this species as a novel experimental system for studying extreme environmental adaptation mechanisms in insects, providing a theoretical foundation for optimizing sexual aphid conservation protocols in gallnut cultivation systems. However, this study did not delineate the molecular architecture of respiratory metabolic pathway regulation in S. chinensis—a critical knowledge gap that could be elucidated through integrated metabolomic profiling and flux analysis. Furthermore, the influencesn of environmental factors (such as thermal and hygric)on the RMR of sexuales require systematic evaluation via controlled laboratory simulations, which would enable refinement of environmental parameter optimization in artificial rearing systems.” (lines 305-313).

Round 2

Reviewer 1 Report

Comments and Suggestions for Authors

I appreciate the authors effort and I consider, that my comments were fully followed. Now, the paper is suitable for publication in its present form.

Author Response

I appreciate the authors effort and I consider, that my comments were fully followed. Now, the paper is suitable for publication in its present form.

We sincerely thank you for your constructive comments and for confirming that our revisions have fully addressed your concerns. We are grateful for your final assessment that the manuscript is now suitable for publication.

The English is fine and does not require any improvement.

Thank you for your kind remarks regarding the English language. Your confirmation that no improvements are needed is gratifying.

Reviewer 2 Report

Comments and Suggestions for Authors

The manuscript “Intergenerational and intersexual differentiation in respiratory metabolic rates of Schlechtendalia chinensis: a comparison across sexuales, parental sexuparae, and progeny fundatrices” (formerly “Analysis of respiratory metabolic rates across three distinct generations of gall aphid, Schlechtendalia chinensis – the sexual generation and its parental and progeny generations” (authors: Shuxia Shao, Bo Jiang, Xin Xu, Zhaohui Shi, Chang Tong and Zixiang Yang) has been significantly revised by the authors and is now, in my opinion, quite ready for publication.

Author Response

The manuscript “Intergenerational and intersexual differentiation in respiratory metabolic rates of Schlechtendalia chinensis: a comparison across sexuales, parental sexuparae, and progeny fundatrices” (formerly “Analysis of respiratory metabolic rates across three distinct generations of gall aphid, Schlechtendalia chinensis – the sexual generation and its parental and progeny generations” (authors: Shuxia Shao, Bo Jiang, Xin Xu, Zhaohui Shi, Chang Tong and Zixiang Yang) has been significantly revised by the authors and is now, in my opinion, quite ready for publication.

Thank you very much for your positive evaluation of our manuscript. We are delighted to hear that you believe it is now quite ready for publication after the significant revisions we have made.

The English is fine and does not require any improvement

Thank you for your kind remarks regarding the English language. Your confirmation that no improvements are needed is gratifying.